# Crowd Monitoring and Localization Using Deep Convolutional Neural Network: A Review

**Akbar Khan** 1, **Jawad Ali Shah** 1,*, **Kushsairy Kadir** 1, **Waleed Albattah** 2 **and Faizullah Khan** 3

1   Electronic Section, Universiti Kuala Lumpur British Malaysian Institute, Selangor 53100,  Malaysia; akbar.khan@s.unikl.edu.my (A.K.); kushsairy@unikl.edu.my (K.K.)
2   Department of Information Technology, College of Computer, Qassim University, 51921 Buraydah, Saudi Arabia; w.albattah@qu.edu.sa
3   Department of Telecommunication Engineering, Balochistan University of Information Technology, Engineering and Management Sciences, Quetta 87300, Pakistan; faizullah.khan@buitms.edu.pk
*   Correspondence: jawad@unikl.edu.my

**Abstract:** Crowd management and monitoring is crucial for maintaining public safety and is an important research topic. Developing a robust crowd monitoring system (CMS) is a challenging task as it involves addressing many key issues such as density variation, irregular distribution of objects, occlusions, pose estimation, etc. Crowd gathering at various places like hospitals, parks, stadiums, airports, cultural and religious points are usually monitored by Close Circuit Television (CCTV) cameras. The drawbacks of CCTV cameras are: limited area coverage, installation problems, movability, high power consumption and constant monitoring by the operators. Therefore, many researchers have turned towards computer vision and machine learning that have overcome these issues by minimizing the need of human involvement. This review is aimed to categorize, analyze as well as provide the latest development and performance evolution in crowd monitoring using different machine learning techniques and methods that are published in journals and conferences over the past five years.

**Keywords:** crowd monitoring; crowd counting; crowd density estimation; deep convolutional neural networks (DCNN); crowd behavior

## 1. Introduction

Crowd is same or different set of people arranged in one group and motivated by common goals. There are two types of crowd namely structured crowd and unstructured crowd. In the former, the direction of the movement is towards a common point and people are not in scattered form while in the later type the direction of the people is not towards a common point and they are usually in scattered form [1].

Crowd monitoring has a wide range of applications such as, safety monitoring, disaster management, traffic monitoring and design of public spaces. This variety of applications have encouraged researchers throughout numerous fields to develop models for crowd monitoring and associated tasks such as counting [2–5], density estimation [6–8], tracking [9], scene understanding [10], localization [11] and behavior detection [12,13]. Among these, the crowd counting and density estimation are important tasks and represent fundamental building blocks for several other applications [14]. There are three most commonly used methods for crowd counting namely, object detection based counting [15–17], clustered based counting [18] and regression based counting [19,20]. In object detection based counting; the object detectors are trained to localize the position of every person in the crowd for counting. Cluster based crowd counting consists of identifying and tracking visual features. Feature trajectories that show coherent motion are clustered and the numbers of cluster is regarded as a estimation of moving objects [21]. Regression

based counting estimates the crowd count by carrying out regression between the image features and crowd size. Though the regression based methods are working good in situations of high density as they capture generalized density information from the image of crowd, still it has two main limitations i.e., performance degradation due to overestimating of count in low density situations and improper distribution of crowd in the scene [22]. One may consider that conventional crowd counting methods such as patch based approach, object based approach and rich feature approach [8,23,24], which depend on either detection or regression, are limited when handling real scene with unavoidable density variations. However, these methods can only be used to estimate and count the people in low density situations in which all parts of the people are fully visible. The performance of these methods deteriorates when applied in high density situations. An ideal counting method should have the adaptive ability to choose the appropriate counting mode according to crowd density [20]. Exact crowd counting and localization are indispensable for handling high density crowds. Localization means to get the accurate location of the heads in an image. Head is the only visible part through which localization can be found in highly dense crowd images or videos [25]. Although numerous steps have been taken in the detection of human heads [16,17,26], head detection is still a challenging task. As a result of the variation in the scale appearance of heads, it is still a big problem to exactly discriminate human heads from the background. For localization in crowded scene, density map has been used as a regularizer during the detection [27]. In computer vision, crowd behavior detection in video surveillance is one of the latest research areas [1]. Crowd behavior detection has many application domains such as automatic detection of riots or chaotic acts in crowd and localization of abnormal regions [28]. Detection of crowd behavior is extensively used in order to monitor and maintain the surveillance of public places like sports events, markets, religious and political gatherings, etc. There are two categories of crowd behavior detection namely global crowd behavior detection and local crowd behavior detection [29]. In global crowd behavior detection a large area is affected, while in the local crowd behavior detection it affects the limited area of the crowd and the behavior of an individual is quite different from its neighbor. Multi scale texture analysis is used for assessing the behavior of crowd in video sequences [30]. The aim of video surveillance is used to detect abnormal human behavior [31].

### 1.1. Rational

Reviewing the literature is one of the most important activities in research. This paper is the first paper in a series of research papers in the field of crowd monitoring. According to our plan of current working on a substantial crowd monitoring project funded by Ministry of Education in Saudi Arabia, it is a very crucial phase to study the literature and analyze it in order to address different aspects of the subject under investigations. Any contribution in any subject or field cannot be achieved with a considerable knowledge of the state of the art. We believe this paper will provide us as well as interested researchers with overview of existing studies in the field of crowd monitoring and management. We also expect this review provides a novel synthesis of the current research works, which we hope can lead to a new means of considering crowd monitoring as well as finding any available gaps.

### 1.2. Datasets

Various datasets containing crowd videos and images are publicly available and are being used to validate the experimental results. Some of the publicly available datasets along with its description are shown in Table 1.

UCSD dataset was the first dataset used for people counting [32]. The data has obtained through a camera installed on a pedestrian pathway. The dataset includes 2000 frames 238 × 158 of video sequences, along with ground truth annotation of each pedestrian in every fifth frame having 49,885 pedestrian in total. The Mall dataset has been collected by means of surveillance cameras installed in a shopping mall [33]. It has a total of 2000 frames with size of 320 × 240. UCF_CC_50 dataset [2] is a challenging dataset comprising of a wide variety of densities and various scenes. This dataset has been obtained from different places like concerts, political protests, stadiums and marathons.

The entire numbers of annotated images are 50 containing 1279 individuals per image. This dataset has a varying resolution and the individuals differ from 94 to 4543 representing a large variation in the image. The drawback of this type of dataset is that, there is only limited number of images available for training and evaluation. WorldExpo'10 dataset introduced in [34] has been used for cross scene crowd counting. The dataset comprises of 3980 frames of size $576 \times 720$ with 199,923 labeled pedestrians. The maximum crowd count through this dataset is limited to 220 and is insufficient for evaluating extremely dense crowds counting. The Shanghai Tech dataset [35] has been introduced for large scale crowd counting containing of 1198 images with 330,165 annotated heads. In terms of annotated heads, this dataset is one of the largest. The dataset contains two types, namely Part A, Part B. Part A is made of 482 images taken from the internet randomly. Whereas Part B comprises of 716 images collected from the metropolitan street in shanghai. The most recent dataset is UCF-QNRF [11] having 1535 images. Within this dataset, the number of people fluctuates from 49 to 12,865 making a massive variation in density. Furthermore, it has a huge image resolution ranging from $400 \times 300$ to $9000 \times 6000$ and consists of crowd videos with varying densities and perspective scales. CUHK dataset has been collected from diverse locations namely, street, shopping malls, airports and parks. The dataset comprises of 474 videos clips from 215 scenes [36].

**Table 1.** Description of datasets.

| Datasets | Description | No.of Images | Resolution | Min | Ave | Max | Overall Count | Accessibility |
|---|---|---|---|---|---|---|---|---|
| UCSD | People counting | 2000 | $238 \times 158$ | 11 | 25 | 46 | 49,885 | Yes |
| MALL | People counting | 2000 | $320 \times 240$ | 13 | - | 53 | 62,325 | Yes |
| UCF_CC_50 | Density estimation | 50 | Variable | 94 | 1279 | 4543 | 63,974 | Yes |
| WorldExpo'10 | Cross scene crowd counting | 3980 | $576 \times 720$ | 1 | 50 | 253 | 199,923 | Yes |
| Shanghai Tech A,B | Crowd counting | 482 | Variable, $768 \times 1024$ | 33 | 501 | 3139 | 241,677 | Yes |
| UCF-QNRF | Crowd counting and localization | 716 | $400 \times 300$ to $9000 \times 6000$ | 9 | 123 | 578 | 88,488 | Yes |
| CUHK | Crowd behavior | 1535 | Variable | 49 | 815 | 12,865 | - | Yes |

This review paper is mainly focusing on the crowd monitoring (crowd counting, crowd localization and behavior detection). The rest of the paper is organized as follows. Section 2 shows search methodology and Taxonomy Level. In Section 3, crowd monitoring approaches taken from the previous literature are summarized in tabular form. Section 4 shows crowd related research approaches. Section 5 shows the convolutional neural network and deep CNN frameworks. Section 6 represents the discussion and Section 7 elaborates conclusions.

## 2. Search Methodology and Taxonomy Level

Figure 1 shows the growth of published papers on crowd monitoring (Crowd counting, localization and behavior) using different machine learning methods and techniques. The graphs show the two main databases of scopus and web of science in which the research papers have been published from year 2014 to 2019. We have searched crowd monitoring keyword in both databases and found different papers with different methods and techniques. The details of the papers published in both databases are shown in graphical view as shown in Figure 1.

*Taxonomy Level*

The entire taxonomy level of crowd monitoring has been shown in the form of flow chart. Basically the crowd related research approaches have been categorized into two domains based on the literature review namely crowd management and crowd monitoring. Then we have made categories of crowd monitoring i.e., counting, localization and behavior. Focusing on crowd monitoring, we just review the

crowd management and did not categorize it further. After that, the counting section has been divided into two parts namely density estimation and people counting. In localization there are three main sub categories i.e., counting and localization, estimation and localization and anomaly detection and localization. Finally, the behavior category has been divided into three sub categories i.e., individual behavior estimation, anomalous behavior detection and normal and abnormal behavior detection. The taxonomy level of crowd monitoring (crowd counting, localization and behavior) has been shown in Figure 2.

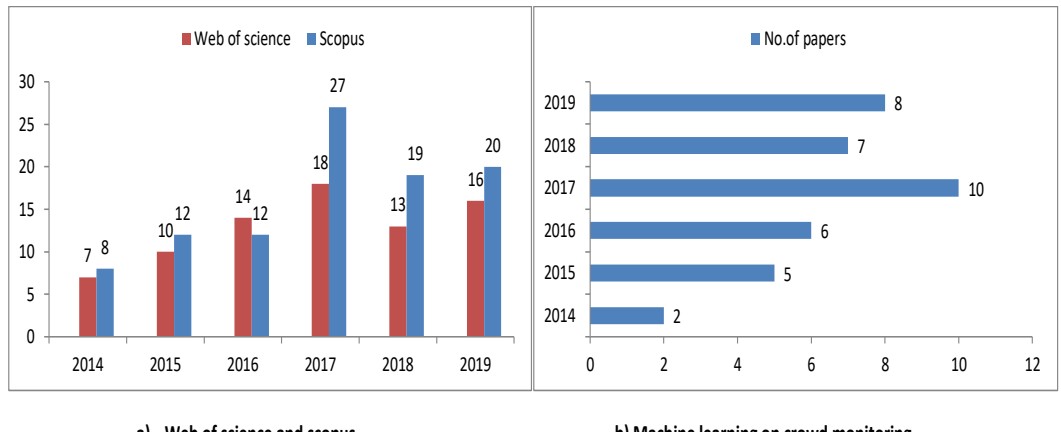

a)   Web of science and scopus                b) Machine learning on crowd monitoring

**Figure 1.** (**a**) The list of research papers published in web of science and scopus on crowd monitoring from 2014–2019 while (**b**) shows machine learning on crowd monitoring

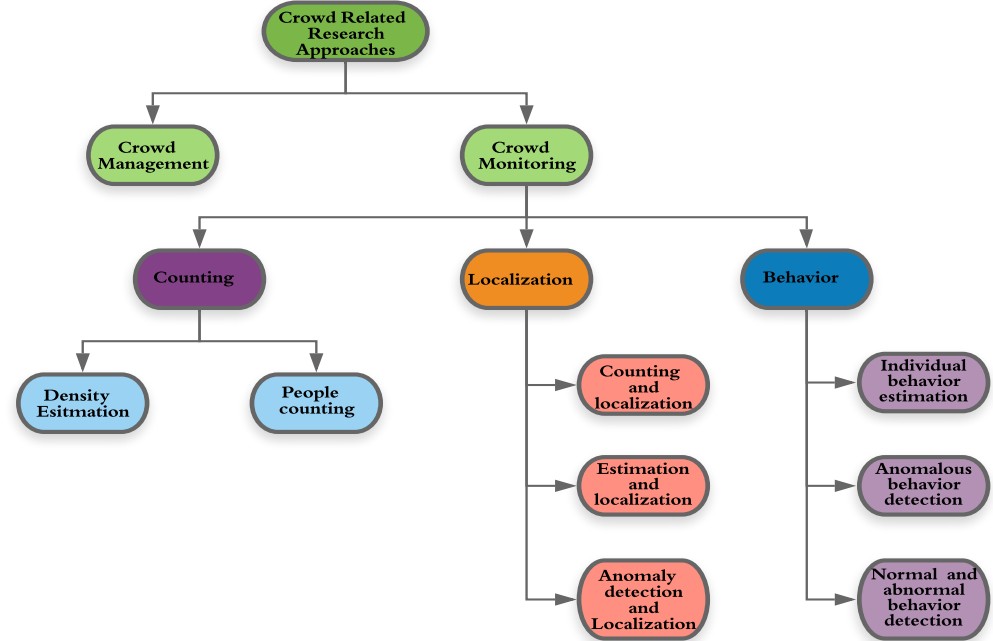

**Figure 2.** Crowd related research approaches.

## 3. Crowd Monitoring (CM) Approaches

Table 2 shows completely about all the research papers about crowd counting, localization and behavior detection. Different research papers with a keyword "crowd monitoring" have been searched while using two mainly used databases i.e., web of science and scopus from 2014–2019, respectively. We have found different methods and techniques related to crowd monitoring and tried to review it completely. Further the elaboration of the crowd counting, localization and behavior have been deliberately discussed in tabular form as shown in Table 2.

**Table 2.** Crowd Monitoring (CM) approaches.

| Ref | Process | Frameworks/Methods | Performance | Conclusion/Result |
|---|---|---|---|---|
| [4] | Crowd monitoring (Counting) | Combination of crowd size estimation and counting | Accuracy 90% | Classification with MSE 0.0081 |
| [37] | Crowd monitoring (Counting) | Neural network and regression trees using fisheye camera | BPNN provides the best estimation | BPNN can deal 9 frames in second |
| [38] | Crowd monitoring (Estimation) | ICrowd framework was designed on a three-layer approach, device layer, middleware layer and the application layer | No experimental results | Capable of location updates |
| [39] | Density estimation (Estimation) | Airborne camera systems, support vector machine and Gabor filter | Depends on good training samples and similar images | Gabor filter plays a prominent role in real scenes of images |
| [40] | Crowd monitoring (Controlling crowd movement) | Information management module and decision support system | Expert system module performs well | Real-time crowd density measurements and communications during hajj |
| [41] | Crowd Counting (Normal/abnormal event) | Median filter and Kalman filter | Accuracy 95.5% | Robust smart surveillance system |
| [11] | Estimation and localization(Density map) | Deep CNN networks | Specificity 75.8% | Decrease error rate |
| [42] | Detection and localization (Anomaly detection) | Global and local descriptors, with two classifiers were proposed | Accuracy 99.6% | Achieved good and competing methods with low computational complexity |
| [43] | Counting and Localization (Human heads) | CNN | DISAM outperforms for UCSD and WorldExpo'10 datasets with the lowest MAE of 1.01 and 8.65, respectively | Reduction of classification time |
| [25] | Counting and localization | SD-CNN Model | Average Precision and average recall rate 73.58 and71.68 | Reduction of classification time and improvement in detection accuracy |
| [44] | Crowd monitoring(Behavior) | EHCAF | 99.1% accuracy and FNR of 2.8%, | Highly accurate and low FNR |
| [45] | Crowd monitoring (Behavior detection) | Isometric mapping (ISOMAP) | Reduced feature space | Reduction of computation time |

**Table 2.** *Cont.*

| Ref | Process | Frameworks/Methods | Performance | Conclusion/Result |
|---|---|---|---|---|
| [46] | Crowd behavior analysis(Behavior) | Spatio-temporal model | Accuracy 98% and 88% | Visual descriptors have extracted and considered for both individual and interactive behaviors |
| [47] | Crowd evacuation (Evacuation behavior) | Legion Evac software | Correlation scores were positive | Reduced evacuation time |
| [48] | Detection of anomaly (Normal and abnormal) | Optical flow and Horn Schunck algorithm | Computation of distance between centroids | A novel approach of abnormal event detection has proposed |
| [49] | Crowd behavior detection (Identify behavior) | Spatio-Temporal Texture model | The STT method demonstrates comparable results of Spatio-temporal Compositions (STC) and Inference by Composition (IBC) | Crowd anomaly detection framework was introduced |
| [50] | Crowd behavior detection(Behavior) | Approximate median filter and foreground segmentation algorithm | Lower false rate | A robust unsupervised abnormal crowd behavior detection has achieved |
| [51] | Violent behavior detection | Hybrid random matrix (HRM) and deep neural network | Accuracy 90.17% and 91.61% | Combining compressive sensing and deep learning to identify violent crowd behavior |
| [52] | Crowd behavior detection | Holistic approach | The performance of this methods yields better results | Holistic approach for abnormal crowd behavior detection has proposed |
| [1] | Crowd behavior detection (Real time) | Scale-invariant feature transform (SIFT) | Accuracy 95% | The combination of SIFT and genetic algorithm has achieved better simulation results |
| [53] | Crowd behavior monitoring (Event detection) | Fixed-width clustering algorithm and YOLO | Accuracy is between 80%-95.7% | The approach has a superior performance on six videos |
| [54] | Abnormal behavior detection(Abnormality) | Optical flow method and SVM | 87.4% accuracy | Higher detection rate for anomaly |

## 4. Crowd Related Research Approaches

After going through an important number of papers, it has been determined that the overall work can be divided into two categories, namely crowd managing and crowd monitoring (counting, localization and behavior). Here is a brief description:

### 4.1. Crowd Management

Crowd management has made enormous progress over the last few years. Within the literature, various models have been proposed. Like in [37], the authors have proposed a Finite State Machine (FSM) model to simulate the movement of crowd during Tawaf (to move around the Kaaba seven times as part of the hajj in Mecca). The model can be used to monitor and manage crowd in Mataf (place of Tawaf) in real time. Similarly in [38], the authors have proposed a framework weighted round robin to overcome the congestion and overcrowded during hajj (pilgrimage). The framework was designed to be proactive in accurately predicting potential problems through the use of smart monitoring of each path of rituals locations. A model has planned to count the number of people by using non-participatory (non-intrusive) technique supported by statistical approach by Design of Experiments (DOE) for crowd safety and management [39]. In [40], an information management module and decision support system was used to monitor and manage the crowd. The proposed framework provides an automated approach for detecting and evaluating the video scene and classifying crowds and traffic management [55].

### 4.2. Crowd Monitoring

Crowd monitoring can further be categorized into crowd counting, crowd localization and crowd behavior.

#### 4.2.1. Crowd Counting

Counting means specifically count the number of people in the crowd. The crowd counting has been discussed by many authors in the literature. Khan et al. [43] proposes a novel head counting and localization technique, Density Independent and Scale Aware Model (DISAM), that performs well for high density crowed where human head is the only visible part in the images. CNN is first used as head detector and later for computing response matrix from the scale aware head proposals to obtain the probabilities of head in the images. In [56], you only look once (YOLO) is a detection technique which is broadly used for the detection of objects in an image with high level of perspective values i.e., maximum threshold value. Xu et al. [57] recommended CNN and learn to scale that generate multi polar normalized density maps for crowd counting. It extracts a patch-level density map by a process of density estimation and clusters then into multiple levels of density. Each patch density map is normalized via an online learning strategy for the center with multi polar loss. In [58] the crowd density of surveillance videos is measured using CNN and short term memory. Two classical deep convolutional networks namely Googlenet [59] and VGGnet [5] were used for estimating crowd density [60]. Similarly, [4] first approximate crowd size estimation and secondly count the exact number of people in the crowd. The efficiency remains unchanged in the terms of its accuracy of (90%).

#### 4.2.2. Crowd Localization

Localization of crowds in crowded images received less attention from the research community. With localization information, one can figure out how people are distributed in the area, which is very important for crowd managers [43]. Information about localization can be used to detect and monitor an individual in dense crowds [61]. In order to identify the location of head in an image a regression guided detection network (RDNet) has proposed for RGB-Datasets that can simultaneously estimate head counts and localize heads with bounding boxes [62]. Similarly in [27], a density map has been used to localize the heads in dense image with accurate results. In [63], localization has been identified

while using LSC-CNN with the help of a metric named as Mean Localization Error (MLE). This model has achieved a remarkable performance in terms of its accuracy. Compressed Sensing based Output Encoding (CSOE) has been proposed which can help to improve the efficiency of localization in highly dense crowded situation [64].

### 4.2.3. Crowd Behavior

Crowd behavior analysis and detection have become a primary part everywhere for peaceful event organization [65]. The difficulties of behavior identification and abnormal behavior detecting are very important issues in video processing [66]. The researchers have proposed different methods and techniques for the crowd behavior detection. Some of the close related works are elaborated here. In [66,67], image processing with optical flow and motion history image techniques were used to detect the behavior of crowd. Similarly in [54], an optical flow method with Support Vector Machine (SVM) was used for abnormal behavior detection. In [68], Cascade Deep AutoEncoder (CDA) and with the combination of multi-frame optical flow information have been proposed for the detection of crowd behavior. Isometric Mapping (ISOMAP) [45], spatio- temporal [46] and spatio-temporal texture [49] models were used to detect the anomalous crowd detection. In [51] Hybrid Random Matrix (HRM) and deep neural network were used for the detection of violent behavior detection. One uses SIFT feature extraction technique [1] and Fixed-width clustering algorithm with YOLO were used to detect crowd behavior [53].

## 5. CNN and Deep CNN Frameworks

Deep CNNs are special types of Artificial Neural Networks (ANNs) that learn hierarchical representation from the spatial information contained in digital images. It was originally design to process multidimensional (2D and 3D) arrays of high resolution input datasets such as images and videos [69–71]. The first deep CNN architecture was AlexNet [69] having seven hidden layers with millions of parameters. Deep convolutional neural networks have achieved great success on image classification [70], object detection [71], crowd counting [72] and people localization [73]. The brief structure of deep CNN is shown in Figure 3.

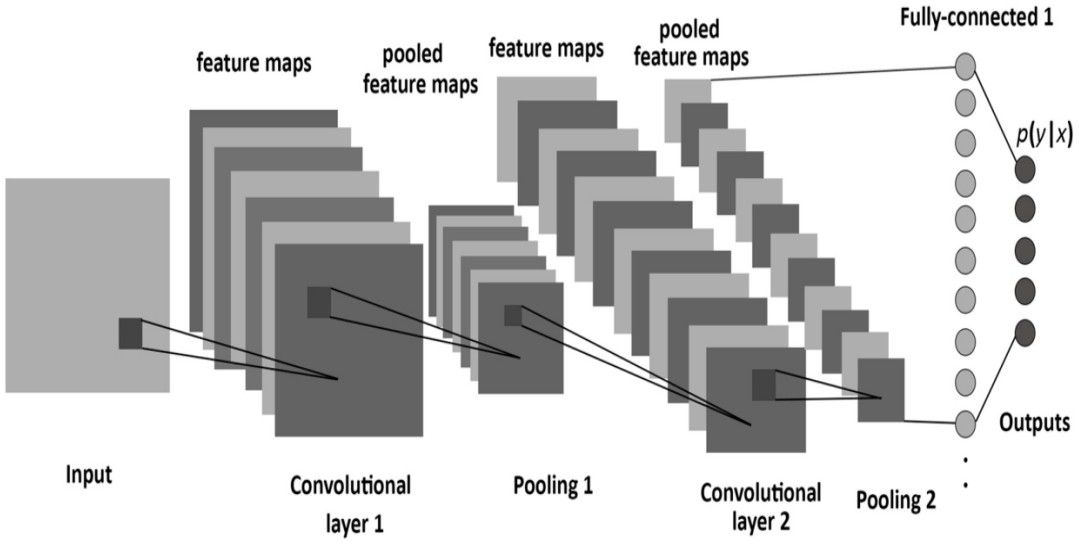

**Figure 3.** Structure of deep convolutional neural networks (CNN) [74].

The success of convolutional neural networks (CNN) and deep convolutional neural networks (DCNN) in various computer vision tasks has inspired researchers to leverage their ability to learn

nonlinear functions from crowd images to their respective density maps or counts [75]. A variety of CNNs and DCNNs have been proposed in the literature which are used for crowd monitoring. Recent studies are included in Table 3 which represents different methods and techniques used for crowd monitoring i.e., crowd counting, density estimation and localization. The selection of research papers is from 2017-2019 respectively in which deep convolutional neural networks, scale driven convolutional networks and simple convolutional neural networks have been used. The experimental results were evaluated using different datasets namely UCSD [32], world expo'10 [19], UCF-CC-50 [2], Shanghai Tech Part A,B [35], UCF-QNRF [11] and subway-carriage [60] datasets. Further details are presented in Table 3.

**Table 3.** CNN and deep CNN

| Ref | Crowd Monitoring | Methods/ Techniques | Datasets | Research focus | Accuracy |
|-----|-----------------|---------------------|----------|----------------|----------|
| [25] | People counting and localization | SD-CNN model | UCSD, world Expo'10 and UCF-CC-50 | Count and localize | Reduced the classification time |
| [43] | Crowd counting and Localization | DISAM | UCSD and World expo'10 | Detection, estimation and localization | Reduced the classification time |
| [57] | Crowd counting | SPN+L2SM | ShanghaiTech A,B, UCF-CC-50 and UCF-QNRF | Large variation in density for crowd counting | 4.2%, 14.3%, 27.1% and 20.1% rates of MAE |
| [11] | Estimation and localization | Composition loss | Shanghai Tech Part A, Shanghai Tech B, UCF-CC-50 and UCF-QNRF data sets | Counting, estimation of density map and localization. | Decreased the error rate of compositional loss. |
| [60] | Crowd managing and monitoring | Deep CNN | Subway-carriage scenes | Crowd density estimation | 91.73% |

## 6. Discussion

This section describes the pairwise comparison of various methods and test datasets. For comparison we have selected some state of the art models which are widely used for crowd monitoring. Table 4 contains the summary of pairwise comparison using MAE and MSE as benchmarking parameters. The CNN based crowd counting and localization algorithms presented in [8,22,34,35,76] are compared with Scale Driven Convolutional Neural Network (SD-CNN) using UCSD, UCF-CC-50, WorldExpo'10 and ShanghaiTech Part A, B datasets. The MAE and MSE of SD-CNN are lesser than those of other models on UCSD, UCF-CC-50 and WorldExpo'10 datasets, respectively. Comparing these models, SD-CNN has the ability to count and localize the human heads in both low and high level density crowd images. Similarly, Density Independent and Scale Aware Model (DISAM) has the lowest MAE as compared to other models tested on UCSD and WorldExpo'10 datasets. Unlike previous models which can only count the people in dense crowd, DISAM has the ability to handle both counting and localizing people in the dense crowd. Finally, we have compared SD-CNN with DISAM on different datasets and concluded that SD-CNN has lower rate of MSE on UCSD dataset and MAE on WorldExpo'10, respectively.

In crowd monitoring and counting problems, researchers try to explore the domain and have applied different machine learning techniques and methodologies to count and localize the crowd as well as their anomalous behavior. In [4] the authors have presented the approximate size estimation and counting the accurate number of people in the crowd. The model has achieved 90% accuracy and it has been shown that the efficiency is not affected by increasing the number of people. The Kalman filtering approach and KL-divergence technique have been used [77] to monitor and count the crowd in smart city. Similarly to monitor and estimate crowd density, deep convolutional network has been used in [60]. There are three different classifiers namely multiple linear regressions, back propagation neural network and regression trees which have been applied and used in [78] to monitor and count the number of people in dense crowd. In [79] the proposed model has been divided into two folds;

firstly, to propose density estimation of the crowd and secondly, to count the number of people using K-Gaussian Mixture Model (GMM). A new model has been proposed namely identifiable crowd monitoring (iCrowd) using three layers approach i.e., device layer, middleware layer and application layer to identify and monitor the crowd [80]. A Feature From Accelerated Segment Test (FAST) algorithm is introduced in [81] to detect and estimate the number of people in a crowd. The deep learning frameworks like CNN and Long Short Term Memory (LSTM) have been used in [58] for crowd density estimation. In [38] an expert crowd control and management system for hajj has been used with three strategies i.e., to address congestion and overcrowded situation using; First In First Out (FIFO), priority queuing and Weighted Round Robin (WRR). An automatic multiple human detection method using hybrid adaptive Gaussian mixture model was introduced in [82] for human detection. The efficiency of proposed method has further evaluated and analyzed by using Receiver Operating/Output Characteristics (ROC), Mean Absolute Error (MAE) and Mean Relative Error (MRE). The proposed method has shown better results. The crowd can also be monitored and counted by using mobile phones and adopting clustering methods. Through these methods the model has achieved 92% accuracy [83]. The airborne camera systems with the techniques of Support Vector Machine (SVM) and Gabor filters have also been used in [84] for crowd monitoring and density estimation. The quality of results clearly depends upon good training samples and similar images. In [40], the authors have developed a decision support system and information management module for the real time crowd density measurements. This model has also been implemented for the crowd monitoring during Hajj. In [44], the authors have presented a framework for crowd behavior using an Enhanced Context-Aware Framework and achieved experimental results with the accuracy of 99.1% and 2.8% of False Negative Rate (FNR) indicating a significant improvement over the 92.0% accuracy and FNR of 31.3% of the Basic Context-Aware Framework (BCF). The detection of anomalous crowd behavior has been monitored in [45] using Isometric Mapping (ISOMAP). During the monitoring and detecting of crowd behavior the ISOMAP has reduced the computational time significantly. To quantify the crowd behavior analysis a spatio-temporal model has been proposed in [46] on CUHK and UMN data sets. The accuracy achieved for both data sets are 98% and 88%, respectively. A software named as legion Evac has been used in [47] for the behavior of crowd evacuation, during simulating legion Evac calculates various metrics that reflect a holistic pattern of crowd evacuation, which capture the behavior of crowd. In [85], authors have developed a probabilistic detection of crowd events (running, walking, splitting, merging, local dispersion and evacuation) on Optical Flow Manifolds (OFM) using Optical Flow Vector (OFV) and Optical Flow Bundles (OFB). Dealing with the issue of different behaviors captured in surveillance video for the use of normal and abnormal behavioral detection, clustering based group analysis has been used in [48] and described certain group behavior, such as collectivity, uniformity and conflict. In [49], the authors have proposed a Spatial Temporal Texture (STT) model which can automate and identify crowd behavior under complex real life situation. The anticipated STT method demonstrates similar results of Spatio-Temporal Composition (STC) and Inference by Composition (IBC) and used less time and a smaller amount of system memory resources. In [50], the authors have presented an unsupervised abnormal crowd behavior detection using approximate median filter and foreground segmentation algorithms. In [51] the authors have proposed a model that may detect and identify the violent behavior of crowd using hybrid random matrix and deep neural network. In [52], the authors presented a novel method for detecting crowd behavior in video sequences using probability model of speed and direction. This method comprises of two main phases; building the motion model (speed and direction) and comparing the model of different frames to detect anomalies. A Scale Invariant Feature Transform (SIFT) technique is used in [1] for detection of crowd behavior in real time video sequences. Similarly, a fixed width clustering algorithm and YOLO have been used in [53] to detect the crowd behavior in video surveillance. In [54] the authors have suggested an effective and concrete method for detecting abnormalities on the basis of optical flow path of the joint points for each human body. The method has an expressively higher detection rate on the public data set with 87.4% accuracy. In [41], the authors have presented a unique multi

person tracking system for crowd counting and normal/abnormal indoor and outdoor monitoring system using median and Kalman filters, and have obtained 95.5% accuracy in event detection. In [42] the authors proposed a system for identification and localization of anomalies in crowded sense in real time. The performance was calculated on the basis of its accuracy i.e., 99.6%. Similarly, in [43] the authors have proposed a novel model namely Density Independent and Scale Aware model for crowd counting and localization in highly dense crowd and evaluated the model on Mean Absolute Error (MAE). In [25] the Scale Driven Convolutional Neural Network model has proposed to count and localize the crowd. This strategy reduced the classification time significantly and improved the detection accuracy. There are many common problems in research related to crowd monitoring such as scale variation, complex background, localization, etc. which need to be solved by using different techniques. For scale variation, SD-CNN has been proposed in literature with the assumption that the head is the only visible feature in the crowd. In a dense crowd, the issue of scale variation has been addressed by generating a scale aware object proposal. Similarly, for large density variation, learning to scale model has also been proposed. Localization of objects in complex background is still a challenging task. DISAM can be used that has the ability to precisely localize the heads in complex scenes. Localization performance is primarily affected by changing the threshold value, so finding an optimum strategy for this issue is a new direction of research.

**Table 4.** Comparison of surveyed methods and test datasets.

| Ref | UCSD | | UCF-CC-50 | | WorldExpo'10 | ShanghaiTech Part A, B | | UCF-QNRF | |
|---|---|---|---|---|---|---|---|---|---|
| | MAE | MSE | MAE | MSE | MAE | MAE | MSE | MAE | MSE |
| [34] | 1.6 | 3.31 | 467 | 498.5 | 12.9 | - | - | - | - |
| [8] | 1.61 | 4.4 | 235.74 | 345.6 | - | - | - | - | - |
| [22] | 1.03 | 1.37 | 302.3 | 411.6 | - | 49.25 | 76.25 | - | - |
| [35] | 1.07 | 1.35 | 377.6 | 509.1 | 11.6 | 68.3 | 107.25 | - | - |
| [76] | 2.89 | 9.25 | - | - | 26.87 | - | - | - | - |
| [25] | 1.01 | 1.28 | 235.74 | 345.6 | 7.42 | - | - | - | - |
| [34] | 1.6 | 3.31 | 467 | 498.5 | 12.9 | - | - | - | - |
| [22] | 1.03 | 1.37 | 302.3 | 411.6 | - | - | - | - | - |
| [35] | 1.07 | 1.35 | 377.6 | 509.1 | 11.6 | 68.3 | 107.25 | - | - |
| [86] | 1.17 | 2.15 | 406.2 | 404 | 14.7 | - | - | - | - |
| [87] | 1.62 | 2.1 | 318.1 | 439.2 | 9.4 | 60.65 | 91.75 | - | - |
| [88] | - | - | 295.8 | 320.9 | 8.86 | 46.85 | 68.25 | - | - |
| [20] | 1.03 | - | - | - | 9.23 | 20.75 | 29.42 | - | - |
| [43] | 1.01 | - | - | - | 8.65 | - | - | - | - |
| [35] | - | - | 377.6 | 509.1 | - | 68.3 | 107.25 | 277 | - |
| [87] | 1.62 | 2.1 | 318.1 | 439.2 | - | 60.65 | 91.75 | 252 | 514 |
| [88] | - | - | 295.8 | 320.9 | 8.86 | 46.85 | 68.25 | - | - |
| [89] | - | - | 322.8 | 397.9 | - | 46.85 | 71.05 | - | - |
| [90] | 1.04 | 1.35 | 291 | 404.6 | 7.5 | 46.45 | 65.05 | - | - |
| [91] | - | - | 279.6 | 388.9 | - | 43.65 | 66.7 | - | - |
| [92] | - | - | 288.4 | 404.7 | 9.1 | 46.1 | 69.15 | - | - |
| [93] | 1.16 | 1.47 | 266.1 | 397.5 | 8.6 | 39.4 | 65.5 | - | - |
| [94] | - | - | 260.9 | 365.5 | 10.3 | 40.25 | 66.65 | - | - |
| [95] | 1.02 | 1.29 | 258.4 | 334.9 | 8.2 | 37.7 | 59.05 | - | - |
| [11] | - | - | - | - | - | - | - | 132 | 191 |
| [57] | - | - | 188.4 | 315.3 | - | 35.7 | 54.75 | 104 | 173.6 |
| [35] | - | - | - | - | - | - | - | 315 | 508 |
| [2] | - | - | - | - | - | - | - | 277 | 426 |
| [87] | 1.62 | 2.1 | 318.1 | 439.2 | 9.4 | 60.65 | 91.75 | 270 | 478 |
| [89] | - | - | 322.8 | 397.9 | 9.23 | 46.85 | 71.05 | 252 | 514 |
| [96] | - | - | - | - | - | - | - | 228 | 445 |
| [97] | - | - | - | - | - | - | - | 190 | 277 |
| [98] | - | - | - | - | - | - | - | 163 | 226 |

## 7. Conclusions

In conclusion, this review paper provides a comprehensive literature review on crowd monitoring using different machine learning techniques and methods. Existing approaches on crowd monitoring were thoroughly reviewed. From this review, we concluded that Scale Driven Convolutional Neural

Network (SD-CNN) and DISAM models are to be considered as novel models for crowd counting and localization in dense crowd images with highest accuracy on different datasets. These models have the applications to detect the visible heads in an image with respect to its scale and density map. Extensive experiments on different datasets demonstrate that these models have achieved a significant improvement over the previous models as explained in the literature review section. The future development of deep CNN on crowd monitoring and localization has different opportunities and challenges.

**Author Contributions:** A.K. and J.A.S. have collected and prepared the data. A.K., K.K. and F.K. have contributed to review and analysis. W.A. has supervised the process of this review. The manuscript was written by A.K. and J.A.S. All authors have read and agreed to the published version of the manuscript.

**Funding:** This research was funded by Ministry of Education in Saudi Arabia through project number QURDO001.

**Acknowledgments:** The authors extend their appreciation to the Deputyship for Research and Innovation, Ministry of Education in Saudi Arabia for funding this research work through the project number QURDO001. Project title: Intelligent Real-Time Crowd Monitoring System Using Unmanned Aerial Vehicle (UAV) Video and Global Positioning Systems (GPS) Data.

**Conflicts of Interest:** The authors declare no conflict of interest.

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
