# Peer review of "Crowd Monitoring and Localization Using Deep Convolutional Neural Network: A Review"

_applsci, doi:10.3390/app10144781_

Round 1

Reviewer 1 Report

The author described a review literature about crowd monitoring and localization methods that are based on deep convolutional neural networks. Within this topic, the authors made a clear overview of the different application domains and described the open source datasets that were used to validate these state-of-art methods. Within this overview, the authors did not elaborate on the specific field coverage of the camera footages, the deployment height and the amount of light, which will have a large impact on the results. Moreover, to apply such algorithms in real world situation, the time to compute and describe an analysis of the risk is very important for crowd management. I would advise the authors to include such an overview where this parameter is included to increase the novelty and the uniqueness of this paper. Furthermore, i would advise the authors to describe very briefly what deep convolutional neural networks are to increase the readiness level. To conclude, i would appreciate any illustrations to indicate the common problems, solutions and the common assumptions that were made by the state of the art algorithms.  

Author Response

We are thankful to the anonymous reviewers for providing valuable feedback to improve our submitted manuscript.

We have incorporated the suggested modifications. The point-wise reply can be found in the attached document.

Regards,

Authors

Reviewer 2 Report

This paper presents a review on deep CNN techniques about crowd monitoring and localization. The authors categorized the techniques into crowd management and crowd monitoring. They employ 7 datasets and compared the listed methods. However, I am very sorry that I cannot give an accept on this paper for the following reasons: (1) The pairwise comparison of the surveyed methods and test datasets is not presented. This pairwise comparison is the key to compare the methods. (2) The number of surveyed works is not enough. Other review papers present more than 100 related works. (3) The discussion on this paper does not present a summary or author's opinion about the surveyed works. I strongly recommend the authors should present more clear opinion about their survey works. (4) The following typos exist. L92. most latest --> most recent or most up-to-date L161. clusteres --> clusters L161. density.Each --> density. Each L163. memory.Two --> memory. Two L186. casecad ?? L278. reviewed.Based --> reviewed. Based L284. section.The --> section. The

Author Response

(The authors gave the same response as above.)

Round 2

Reviewer 2 Report

The authors successfully answered my questions.